# Intrinsic Cardiac Neuromodulation in the Management of Atrial Fibrillation- A Potential Missing Link?

**DOI:** 10.3390/life13020383

**Published:** 2023-01-30

**Authors:** Tolga Aksu, Dhiraj Gupta, Jamario R. Skeete, Henry H. Huang

**Affiliations:** 1Department of Cardiology, Yeditepe University Hospital, Istanbul 34752, Turkey; 2Liverpool Centre for Cardiovascular Science, University of Liverpool and Liverpool Heart and Chest Hospital, Liverpool L14 3PE, UK; 3Department of Cardiology, Rush University Medical Center, Chicago, IL 60612, USA

**Keywords:** cardioneuroablation, atrial fibrillation, ganglionated plexus, autonomic nervous system

## Abstract

Atrial fibrillation (AF) is the most common supraventricular arrhythmia that is linked with higher cardiovascular morbidity and mortality. Recent evidence has demonstrated that catheter-based pulmonary vein isolation (PVI) is not only a viable alternative but may be superior to antiarrhythmic drug therapy for long-term freedom from symptomatic AF episodes, a reduction in the arrhythmia burden, and healthcare resource utilization with a similar risk of adverse events. The intrinsic cardiac autonomic nervous system (ANS) has a significant influence on the structural and electrical milieu, and imbalances in the ANS may contribute to the arrhythmogenesis of AF in some individuals. There is now increasing scientific and clinical interest in various aspects of neuromodulation of intrinsic cardiac ANS, including mapping techniques, ablation methods, and patient selection. In the present review, we aimed to summarize and critically appraise the currently available evidence for the neuromodulation of intrinsic cardiac ANS in AF.

## 1. Introduction

As the most frequent chronic arrhythmia, atrial fibrillation (AF) is an important and growing global health issue, as the majority of patients require lifelong treatment in some form, whether it be rate- or rhythm-control for symptom relief, or consideration for long-term oral anticoagulation to reduce the risk of stroke [1]. In the last several decades, catheter ablation using the technique of pulmonary vein (PV) isolation (PVI) has emerged as a cornerstone treatment strategy for patients with paroxysmal AF, as prior studies have reported that ectopic firing from foci harbored within the PV muscles sleeves are responsible for triggering the majority of episodes in patients with paroxysmal AF cases. Catheter ablation has been shown to be especially reasonable in patients when antiarrhythmic drugs have been ineffective, or are contraindicated or poorly tolerated, but also as a first-line therapy [2,3,4,5]. Although PVI is clinically effective in many patients, a large number of variables, such as the presence of persistent or long-standing persistent AF, comorbidities such as obesity and sleep apnea, the time from the first diagnosis of AF to evaluation for catheter ablation, the definition of success (intermittent monitoring with Holter recording vs continuous monitoring with implantable loop recorders), and the duration of follow-up influence the success rate during follow-up [6,7,8,9,10,11]. 

The intrinsic cardiac autonomic nervous system (ANS) is essential to control normal cardiac physiology and mechanical contraction. After the demonstration that the proclivity for initiation and maintenance of AF itself is partially influenced by intrinsic cardiac ANS [12,13,14,15], neuromodulation of intrinsic cardiac ANS by endocardial catheter ablation of ganglionated plexi (GPs) has been investigated [16,17,18,19]. The present comprehensive review focuses on the topic of neuromodulation of the intrinsic cardiac ANS with the aims of defining (1) our current state of knowledge, (2) potential applications for neuromodulation in appropriately selected patients, (3) different techniques currently available, and (4) a discussion of potential future areas of advancement for clinical applications of this technique.

## 2. Intrinsic Cardiac Neuroanatomy

In the ANS, the fibers traveling from the central nervous system to the ganglia are known as the preganglionic fibers, whereas the fibers bound for the visceral effector organ are the postganglionic fibers. The ganglia of the parasympathetic division of the heart are distributed mostly in the epicardial regions. Histologic studies of cadaver heart sections have revealed an interconnected group of autonomic ganglia, now referred to as GP [20,21,22,23,24]. The following atrial locations were consistently identified in humans and named according to the GP nomenclature proposed by Armour et al. [2] (Figure 1): (1) the posterosuperior surface of the right atrium (RA) adjacent to the superior vena cava (SVC) junction and RA (superior right atrial GP; [RSGP]); (2) the posterosuperior surface of the left atrium (LA) (superior left atrial GP; [LSGP]); (3) the interatrial groove (posterior-inferior right atrial GP; [RIGP]); (4) the posteromedial surface of the LA (posteromedial left atrial GP; [PMLGP]); (5) the posterolateral surface of the LA (posterolateral-inferior left atrial GP; [LIGP]); and (6) the extensions of the RIGP and PMLGP. The ligament of Marshall (LOM) is also densely innervated by the left vagus nerve, with its immunohistochemistry demonstrating predominantly cholinergic nerve bundles, and it may innervate surrounding structures such as the PVs, left atrial appendage (LAA), and musculature of the coronary sinus [25].

Despite this simplistic definition, the autonomic ganglia may have substantial anatomic variations, including the degrees of ganglion density and size, with some GPs discernable to the naked eye, whereas others are only visible with a microscope [26]. Furthermore, the hearts of bigger mammals contain more than one thousand ganglia [26,27]. By staining the intrinsic cardiac ANS on the whole (non-sectioned) heart, Pauza et al. demonstrated that the neuronal control of the heart is provided by one intrinsic epicardiac neural plexus. The postganglionated nerves extend to specific atrial or ventricular regions around the sinoatrial node, the muscle sleeves of the caval veins and PVs, and near the AV node via specific neural elements referred to as the epicardial neural ganglionated subplexi [26,27,28]. Because autonomic ganglia are always found along those subplexal nerves, GP terminology used to describe the most frequent distributions of the intrinsic cardiac ANS might still be valid to define those areas anatomically.

## 3. The Role of the Intrinsic Cardiac Autonomic Nervous System in Atrial Fibrillation

To uncover the potential mechanisms for autonomic-induced AF, different hypotheses have been put forward in animal models. Alterations in autonomic tone are carried out by impulses from these GPs and can induce changes in local cellular electrophysiologic properties, which may responsible for the pathogenesis of AF, although the exact cellular mechanisms have not been fully characterized [29]. In an experimental model, autonomic nerve stimulation was shown to decrease the action potential duration (APD) in the musculature of a canine pulmonary sleeve model and initiate the rapid firing of early after depolarizations. A muscarinic blockade with atropine prevented the shortening of the APD and rapid firing from the PVs [15]. In an open-chest study similarly conducted in a canine model, the infusion of isoproterenol or acetylcholine induced AF in 12% of the canine subjects. The treatment with atropine was more effective than the administration of the beta-blocker agent, propranolol, in preventing the induction of AF, suggesting a critical role for cholinergic tone in the initiation of AF [30]. In another canine study, Po et al. [14] demonstrated that, after thoracotomy injection of para-sympathomimetics into epicardial GP-rich fat pads, there was spontaneous (36%) or easily inducible (64%) sustained AF. Spatial heterogeneity in atrial refractoriness may also play a role in the re-entry-mediated driver of AF. In a study of nine canines, stimulation at the RSGP at the base of the right superior PV (RSPV) led to a gradient of atrial refractoriness and a window of vulnerability for AF, depending on the distance from the GP stimulation site [31]. In human studies, a dispersion of atrial refractoriness based on the proximity to the GP sites has also been demonstrated by mapping during parasympathetic stimulation via epicardially placed wire electrodes [32,33]. Finally, AF itself may lead to autonomic remodeling and contribute to the maintenance of AF. In a rapid atrial pacing model simulating AF, there was a decrease in the effective refractory period, a window of vulnerability, and neural activity of the RSGP during the first 6 h of pacing. The introduction of low-level vagal stimulation during right atrial pacing returned all measures back towards the baseline levels [34]. Autonomic imbalances, such as a derangement in the sympathetic tone, are believed to play an important role in AF in patients with hypertension, obstructive sleep apnea, and heart failure, possibly via cellular, structural, and electrical changes. [35,36,37]. In addition, vagal-mediated AF is a well-described subpopulation of patients who tend to be younger and have paroxysmal forms of AF at the time of presentation. Coumel et al. [38] were the first to report the phenomenon of sinus rate slowing preceding AF in 18 patients as evidence that vagal activity might predispose them to AF initiation. In a study of 77 consecutive patients with paroxysmal AF who wore Holter monitors, the time and frequency domain analyses suggested increases in both vagal and adrenergic tone prior to AF episodes, followed by a period of marked vagal predominance [39]. In a 101-patient study, Mohammadieh et al. [39] demonstrated that patients with paroxysmal AF and obstructive sleep apnea had increased parasympathetic tone and a relative reduction in sympathetic modulation during non-REM sleep when assessed by frequency-domain analysis of heart rate (HR) variability, suggesting vagal predominance as a contributor to AF in the subpopulation. Screening for obstructive sleep apnea is currently recommended in patients for whom rhythm control is sought [37]. Clinical observations of the provocation of AF and experimental evidence of increased ectopy and spatial changes in refractories with adrenergic and cholinergic stimulation, particularly nearby the sleeves of the PVs, have led to the exploration of multiple neuromodulatory therapies aimed to address AF.

## 4. Mapping of Intrinsic Cardiac Autonomic Nervous System for Neuromodulation

The detection of intrinsic cardiac ANS or GPs during an electrophysiological study is fundamental to performing successful catheter-based autonomic neuromodulation. While different approaches have been used to identify important sites of the intrinsic cardiac ANS in atria in the EP laboratory (Table 1), there remains a lack of consensus on the optimal technique for mapping in this regard.

### 4.1. Spectral Analysis

By spectral analysis through the fast Fourier transforms, Pachon et al. [16,40] defined two types of atrial spectral potentials: (1) compact potentials that work like one isolated cell and present homogeneous, fast conduction with a single high-power fundamental frequency and rapid uniformly decreasing harmonics, and (2) fibrillar potentials that are similar to a group of nerve cells and show a heterogenous and low-power fragmented profile with irregular harmonics of high amplitude and wide distribution. The areas containing fibrillar potentials were named “AF nests” by the authors and used to define the localization of intrinsic cardiac ANS or GPs. The investigators found that AF nests were 9.7 times more frequently located in the LA than RA, with the most common locations: the left superior PV insertion (91.1%); interatrial septum (91.1%); RSPV (88.2%); left inferior PV (67.6%); right lateral wall of the RA and crista terminalis (47%); and the insertion of the vena cavae (61.7%).

### 4.2. High-Frequency Stimulation

According to animal experiments, high-frequency electrical stimulation (HFS) of different parts of the LA causes two types of response: (1) autonomic response and (2) normal or nonspecific response. An autonomic response may occur in three different ways: (1) a vagal response (VR) characterized by immediate sinus bradycardia or atrioventricular block; [2] a marked shortening of the atrial RP nearby the stimulated GP; and [3] an initiation of sustained AF, either spontaneously or by a single atrial extrastimulus delivered nearby the GP [13,41,56]. However, HFS at the remaining LA sites did not induce any significant changes in the PR or RR intervals, decrease in atrial RP, or induction of sustained AF with a single atrial extrastimulus [41,56]. Thus, the demonstration of a positive autonomic response may be used to distinguish autonomic innervation sites from uninnervated atrial myocardium. Because each GP has sympathetic and parasympathetic neural elements, autonomic responses to HFS may vary by duration of application, with a tendency for shorter applications to stimulate the parasympathetic fibers. When delivering HFS for longer intervals than 2–5 s, the sympathetic fibers may also be stimulated, potentially blunting the expected parasympathetic response [41,42]. Nakagawa et al. [41] studied the characteristics of autonomic response and electrogram (EGM) morphology during AF and found that sharp, fractionated atrial potentials are found more frequently in adjacent PVs and LA regions nearby stimulated GP. According to sites exhibiting an autonomic response during HFS and fractionated EGM characteristics during AF, five distinct areas were identified: (1) RSGP; (2) RIGP; (3) LSGP; (4) LIGP; and (5) MTGP.

In a recently published study, Kim et al. [43] used a slightly different HFS technique and defined two functional classes of GP: an atrioventricular-dissociating GP type and an ectopy-triggering GP type (ET-GP). A probability atlas of ET-GP revealed a 30–40% probability of ET-GP in the areas of the PV ostia (except for the base of the right inferior PV (RIPV) on the posterior wall), roof, mid-anterior wall, the anterior wall near the RSPV, and the posterior wall near the left inferior PV. Smaller isolated patches of ≥40% probability for ET-GP were confined to the peri-PV region: left PV carina, RSPV antrum, and RSPV ostium on the LA roof [19]. One potential reason for the discrepant findings between Nakagawa and Kim’s studies may be the stimulation by HFS, not only of the epicardial ganglia, but also the nerve extensions within the atrial myocardium from the epicardial ganglia. In a canine-isolated LSPV model, the HFS of axons originating from the GP led to a marked shortening of APD and induction of early after depolarizations and firing from the PV sleeve myocardium. Conversely, the response to the HFS was negated with an infusion of tetrodotoxin. These findings suggest HFS may exert its effects through the stimulation of autonomic axons rather than the electrical stimulation of cardiac myocytes [15,57].

### 4.3. Electrogram Analysis

Based on the compact and fibrillar atrial EGM principles of Pachon et al. [16], Lellouche et al. [58] then analyzed the EGM characteristics based on VRs during RF applications. The EGMs from ablation sites were recorded with a 12-bit analog-to-digital amplifier on 977 samples with a 30–500 Hz bandpass filter. A fractionated atrial EGM during sinus rhythm was defined as an EGM with ≥4 deflections plus a duration of ≥40 ms, as those characteristics best predicted the occurrence of a thermal-induced parasympathetic response with RF ablation. Lellouche further distinguished three main types of LA EGMS in sinus rhythm based on the amplitude (Figure 2): (1) normal EGMs: <4 deflections or <40 ms duration; (2) low-amplitude fractionated EGMs: ≥4 deflections and <0.7 mV amplitude; and (3) high-amplitude fractionated EGMs: ≥4 deflections, ≥0.7 mV amplitude, and ≥40 ms duration. Pachon et al. [16] demonstrated filter settings may have a great impact on the detection of fractioned potentials. The use of 300–500 Hz filters instead of conventional 30–500 Hz filter settings aided in mapping the “fibrillar” myocardium to target for ablation. Thus, in our initial work, the autonomic ganglia sites were detected through a combination of fast Fourier transform analysis of EGMs and HFS [44]. All the EGMs at successful RF ablation sites stimulating an autonomic response demonstrated a fragmented pattern. Based on their superior signal fidelity, the higher high-pass filters improved our appreciation of the EGM fragmentation [16], and so, in subsequent work, our group used 200–500 Hz bandpass filter settings instead of conventional filter settings to target all the fragmented EGMs in the regions during sinus rhythm, which co-localized with the expected anatomic autonomic innervation sites [45]. Indeed, this streamlined electroanatomical mapping-guided approach demonstrated an identical clinical success in comparison to previous combined approaches. Figure 3 the anatomical distribution of GPs in accordance with our definition method.

In a recent study, Kuniewicz et al. [51] attempted to localize fractionated EGMs using a high-density mapping catheter (PentaRay NAV Catheter, Irvine CA) in 35 patients undergoing AF catheter ablation. Using characteristics from Lellouche and our groups [58], the duration, amplitude, and number of deflections were determined for each EGM. In a retrospective analysis of case data, the authors identified predominantly six regions exhibiting fragmented EGMs within the LA. Four regions were in nearby PVs: (1) in front of the RSPV; (2) below the RIPV; (3) LSGP, the roof of the LA, and (4) below the LIPV. Two other regions were (5) the MTGP, located along the LAA-LPV ridge; and (6) the inferoposterior fractionated atrial potentials located directly above the coronary sinus between the RIGP and LIGP on the LA posterior wall. Thus, the identification of fractionated bipolar atrial EGMs with or without the use of high-density mapping catheters may facilitate the rapid and feasible identification of GP sites while shortening the ablation procedure time [48,49]. Unfortunately, the presence of fractionated atrial EGMs may be less specific for GP sites in the presence of a diseased myocardium, such as areas of fibrosis. In a recent study, our group demonstrated that new operators can successfully achieve acute procedural success using a fragmented EGM-guided GP ablation strategy with low learning [50].

### 4.4. Myocardial Innervation Imaging

In 2014, Ben-Haim et al. [59] showed for the first time that 123I-metaiodobenzylguanidine (123I-mIBG), which is internalized by the presynaptic nerve endings of postganglionic neurons, can be used to localize GPs. Twelve patients who underwent mIBG-infusion under a solid-state cardiac camera; contrast-enhanced computed tomography (cCT), or cardiac magnetic resonance imaging (CMR) were automatically co-registered. The LA mIBG-uptake in the epicardial fat pads of the LA was projected on the cCT or CMR with the merged data imported into the 3D electroanatomical mapping system (CARTO 3, Biosense Webster). HFS was then performed at these regions to confirm their correspondence to GP locations before ablation was performed. Utilizing the HFS, all except two sites of focal mIBG uptake were confirmed in five patients who underwent AF ablation. Stirrup et al. [55] defined a high-resolution Cadmium Zinc Telluride camera SPECT/cCT protocol to identify GPs with high accuracy when compared with HFS at those sites. A total of 73 I-mIBG LA-uptake areas were found, of which 59 (81%) identified sites correlated with HFS responsiveness. Thus, SPECT could enhance or eventually replace HFS techniques for the identification of GPs and assist with procedural planning. Moreover, mIBG SPECT also represents an innovative tool to evaluate the extent of LA denervation and dynamics of reinnervation post-PVI. GP ablation may also cause ventricular myocardial denervation in addition to atrial effects. Lemery et al. [60] compared pre-ablation mIBG imaging with early and late imaging post-ablation in five AF patients for whom HFS mapping was also performed. The RF ablation targeted GP antral sites with uptake, in addition to the lesion sets required for PVI. Interestingly, ventricular myocardial denervation was documented in all the patients after the atrial ablation.

### 4.5. Cardiac Computed Tomography

cCT has been used for a long time to increase confidence in performing LA ablation in cases of complex and variable PV and LA anatomy. In a recently published study to define the ability of cCT to identify epicardial adipose tissue for guiding GP ablation, Markman et al. [54] conducted a prospective study of patients who underwent AF catheter ablation. In a total of 15 patients with AF following preprocedural cCT, the atrial anatomy and epicardial adipose tissue near the PVs, coronary sinus, LOM, and SVC-aortic area with attenuation <0 Hounsfield Units were segmented and exported using ADAS software (Galgo Inc). The segmentations were then registered to mapping coordinates (CARTOMerge, Biosense Webster). Fractionated EGMs were identified as ≥4 deflections using 100–500 Hz filter settings. HFS was performed with a definition of VR as >50% R-R interval prolongation. The GP ablation in the study was performed targeting epicardial adipose tissue identified by cCT. An HR increase (>10 beats/min) was observed during the ablation of SVC-aortic epicardial adipose sites and also the right superior epicardial adipose tissue sites in 12 (80%) study patients. An HR decrease (>10 beats/min) was observed when ablating left superior epicardial adipose sites in four (27%) study patients. There was, however, substantial variability in the location and expanse of epicardial adipose tissue regions, with >10 mm variability relative to anatomic landmarks between patients.

### 4.6. Anatomical Approach

An anatomic ablation strategy can be utilized in two different manners: as adjunctive to EGM analysis or HFS [46,51], or both; or it can be utilized as a stand-alone strategy [52]. Although the anatomical distribution of GPs is well demonstrated in animal and human experimental studies, the localization of GPs may demonstrate substantial variability from one patient to another. The advantages and disadvantages of this technique have been summarized in Table 1. Despite the existence of highly specialized techniques, it should be noted that the largest randomized controlled study, thus far, examining the role of adjunctive GP ablation to standard PVI used an anatomical-only approach in the GP ablation group [18]. Furthermore, anatomical ablation (i.e., targeting areas known to host GP in the LA without using surrogate methods for identification of GP) may yield equivalent or even perhaps superior clinical results to methods utilizing HFS for the identification and ablation of GPs in PAF patients [46].

## 5. Ablation Techniques for Neuromodulation of Intrinsic Cardiac Autonomic Nervous System

### 5.1. Endocardial Ganglionated Plexus Ablation

Following the discovery that GP stimulation may result in triggered activity in PVs and fractionated EGMs sites thought to maintain AF, modulating or, in some instances, eliminating key neural connections to the heart by using endocardial ablation techniques has been pursued [18,40,41,46,52,61]. Once GPs are localized during an electrophysiology study by the techniques described above, they may be ablated using radiofrequency (RF) energy. The mapping of GPs should begin with the creation of the anatomic shell of the LA, or the LA and RA, based on the GP mapping and ablation technique. 

In case HFS is being used to map for GPs, the ablation procedure should be performed under general anesthesia rather than conscious sedation. Once positioned at a stable site in the LA, the ablation catheter is paced at a rate higher than the intrinsic sinus rate at a high output at the distal poles to check for the absence of ventricular capture. Typically, at least 80–100 points within a 4–6 mm distance from each other in the LA are globally mapped using HFS [19,41,42,43]. HFS may also be conducted in the RA, adjacent to the SVC and the septum, and also at the IVC near the CS ostium [54]. All the HFS sites with a positive VR are annotated on the three-dimensional electroanatomic map. RF energy (routinely 30–35 W × 30 s but shorter if esophageal heating is obtained) is applied to each positive VR site to the HFS. After each RF application, the HFS should be repeated. If a VR remains, repeat ablations should be attempted until the complete elimination of VR from the thermal response. We have observed that autonomic response characteristics can be variably affected by the patient’s level of sedation [47]. The VR during GP ablation can be categorized according to 3 levels: (1) a > 50% R-R interval increase (level 1); (2) a 20–50% R-R interval increase (level 2); and (3) a < 20% R-R interval increase (level 3). Although ablation of LSGP led to a level 1 VR in 89.6% of cases in the conscious sedation group, level 1 VR was observed in just 22.2% of patients in the deep sedation group (*p* < 0.0001). The proportion of patients with level 1 VR with GP ablation of LIGP was significantly lower in the deep sedation group. For level 2 VR, the ratio of (+) VR during the ablation of LSGP and LIGP was similar between the groups. It should be kept in mind that using an R-R interval change of >20% instead of >50% might improve diagnostic performance when patients are placed under general anesthesia or deep sedation. However, despite its strong pathophysiologic basis, an HFS-based strategy has not yielded a demonstratable advantage over an empirical anatomic ablation approach in AF patients, or those with vasovagal syncope [52,62].

In cases of EGM or spectral analysis-based GP ablation, EGMs with ≥4 deflections or fibrillar potentials at locations anatomically consistent with expected GP sites are then tagged as ablation targets on the operator’s 3D electroanatomic map. Based on the used approach, either both the atria or the LA alone can be included in the analysis. To minimize the diminution of the VR with cumulative GP ablations, after identifying all the target sites in the LA, RF ablation of targeted areas should ideally be performed in the following order: LSGP -> LIGP -> MTGP -> RSGP -> RIGP. Our group has recently demonstrated that each GP site may demonstrate unique neuromodulatory characteristics during RF application from its thermal effects [48]. While ablation of left-sided GPs typically leads to a VR, ablation of the RSGP often increases HR acutely without any observed VR. Further, ablating GP sites in the RA may cause downstream attenuation of the VR ablation during subsequent ablation of the left-sided GPs [49], which is why our laboratory’s preference is to start with the ablation of left-sided GPs. 

Anatomic GP modification may be performed using the technique initially described by Katritsis [52] and later modified by Pokushalov [47,55]. With or without merging cCT, presumed GP clusters can be ablated 1–2 cm outside the PV–LA junction at the following sites by using irrigated-type catheters: LSGP, LIGP, RSGP, and RIGP. The ablation procedure endpoints were the elimination of local electrical activity (peak-to-peak bipolar EGM < 0.1 mV) and the abolishment of any thermal-induced vagal effects with RF applications.

### 5.2. Surgical Ganglionated Plexus Ablation

In the beginning, surgical GP ablation was in addition to the conventional maze procedure and was performed with high success rates (83–93%) [63,64,65]. Because this approach is related to the risks of open-heart surgery, it should only feasibly be considered in patients undergoing valve surgery. Nowadays, surgical GP ablation is performed in addition to PVI, with or without LOM ablation or LAA ligation, by using thoracoscopic approaches [66,67,68,69]. Basically, under general anesthesia, the anterior right GP, which is located in the epicardial fat pad anterior to the right PVs, and the inferior right GP located in the fat pad inferior to the RIPV extending to the inferior posterior wall of the LA, are identified by anatomic landmarks and the VR induced by HFS. On the left side, the LSGPs and LIGPs located in the LA roof fat pad and inferior to the LIPV extending toward the posterior wall of the LA are similarly identified and ablated. Subsequently, the isolation of PVs was attempted by application of bipolar RD energy to clamps that were positioned around the PV antrum. The GPs between the SVC and aorta were not addressed in most studies. HFS is used to check for the effectiveness of ablation. The LOM and LAA can be dissected and ablated based on operator preference. Although early studies showed promising results, the recently published AF Ablation and Autonomic Modulation via Thorascopic Surgery (AFACT) randomized controlled trial did not demonstrate a reduction in AF recurrence at 2 years but was associated with higher adverse events, including rates of major bleeding, SN dysfunction, and also need for pacemaker implantation [69,70].

## 6. Clinical Data for Neuromodulation of Intrinsic Cardiac Autonomic Nervous System in Treatment of Atrial Fibrillation

Approaches to neuromodulation of the cardiac ANS include transcatheter endocardial techniques, thoracoscopic approaches, and open epicardial ablation. The gold standard for AF ablation remains PVI, although GP ablation has been investigated as a standalone approach, and also as an adjunct to PVI, with varying results. The effectiveness of GP ablation when performed by itself has proven to be variable. In a small observational study of 14 patients, Lemery et al. [58] demonstrated that HFS-guided GP ablation alone led to a 50% freedom from AF recurrence at the 8-month follow-up, and Katritsis et al. [59] showed disappointing results (26% freedom from AF) with anatomic guided GP ablation. In two recent randomized studies, HFS-GP ablation alone has shown a somewhat better efficacy of 49–50%, although this was still numerically lower than that with PVI alone at 61–64% [33,34].

Several studies seem to suggest the efficacy of endocardial GP ablation combined with PVI leads to greater efficacy than either technique performed alone. In a 297-patient observational study with a cohort with PAF, Pappone et al. [71] performed the ablation of GP targets with vagal reflexes in addition to conventional PVI with higher 1-year freedom from symptomatic AF recurrences in comparison to PVI only (99% vs 85%) [5]. Katritsis et al. [18] randomized 242 patients with PAF to either PVI only, anatomic-based GP ablation only, or PVI plus anatomic GP ablation. Two-year freedom from recurrent atrial arrhythmia episodes was highest in the PVI plus GP ablation group (PVI alone: 56%; GP ablation alone: 48%; PVI + GP ablation: 74%; *p* = 0.004). No differences in procedural safety events were observed. In another RCT of 264 patients with persistent AF, PVI + GP ablation was seen to be superior to PVI + Linear LA ablation, with higher maintenance of SR at three years (49% vs. 34%; *p* = 0.035) and a lower rate of atrial flutter (6% vs. 18%; *p* = 0.002) [72]. The headline results of all the randomized studies in this area are included in Table 2. Taken together, these studies suggest that endocardial GP ablation may improve the maintenance of SR when performed in addition to PVI.

Surgical approaches for GP ablation have more variable efficacy. The above-mentioned AFACT trial showed no additional benefit from GP and LOM ablation while resulting in an increased need for pacemaker implantation. Similarly, an observational study also found no additional benefit from epicardial GP ablation when combined with cryoablation of the LA during cardiac surgery for maintaining SR with or without AADs [74]. Ultimately, an epicardial approach for GP ablation does not really appear to offer significant advantages for efficacy or safety. Hybrid approaches combining epicardial with endocardial ablation show promise. Although the study was not designed to specifically examine GP ablation, the Convergence of Epicardial and Endocardial Ablation for the Treatment of Symptomatic Persistent AF (CONVERGE) trial is the largest RCT to compare hybrid and endocardial ablation in persistent AF patients [75]. The hybrid ablation arm demonstrated superior freedom from AF than the control arm (67.7% vs. 50%; *p* = 0.036). Whether this was because of the improved durability of posterior wall ablation or better ablation of GPs with epicardial ablation, or both, is unclear [73]. Further evidence is needed to elucidate the impact of hybrid approaches on GP elimination, autonomic modulation, and improved outcomes.

In summary, the published data to date do not yet support neuromodulation of the cardiac ANS system as an alternative to PVI. Whether neuromodulation would result in better outcomes, either in addition to PVI or alone in appropriately selected subgroups of patients such as those with AF recurrence despite effective and persistent PVI, remains unknown.

## 7. Ethanol Infusion in the Vein of Marshall

Because the LOM is both highly cholinergically and adrenergically innervated, its ablation has been proposed as a new target for neuromodulation [76]. As the epicardial structure is inaccessible using conventional ablation techniques in most patients, ethanol infusion has been used for the ablation of this structure. In 2009, Valderabano et al. [77] demonstrated the feasibility of ethanol ablation in a cohort of AF patients. In the recently published randomized VENUS trial, the technique was safely performed in 12 centers [77]. At 12 months, the freedom from atrial arrhythmias was 49.2% in the catheter ablation combined with the vein of Marshall ethanol infusion group in comparison to 38% in the catheter ablation-only group (*p* = 0.04). In the Marshall-PLAN study, a stepwise ablation approach consisting of a vein of Marshall ethanol infusion, PVI, and anatomical isthmuses (mitral, roof, and cavotricuspid isthmus) was tested in patients with persistent AF [78]. At 12 months, 72% of patients were free from atrial arrhythmias after a single ablation procedure without AADs. In those with a full Marshall-PLAN lesion set (n = 68), the 12-month success rate was 79%. These findings confirm the role of the ligament of Marshall for the initiation and maintenance of AF [79]. 

## 8. Unanswered Questions

Several questions remain about the GP detection and ablation modalities: What is the best way of identifying GPs in individual patients that can be done quickly and in an efficient manner? Which ablation method should be selected to effectively attain the elimination of GPs and to avoid potential proarrhythmic consequences of neural plasticity and regeneration that have not yet been well-defined? Based on our understanding from the experimental evidence, a paradoxical interaction between neuromodulation and AF may exist. In a sham study, the ablation of the four major GPs and LOM was compared to a sham control group in canine subjects [80]. In the ablation group, the acute phase demonstrated a significant prolongation of ERP and lower AF inducibility; however, in the chronic phase at 8 weeks post-ablation, the ERP had shortened and AF inducibility was significantly higher in the ablation group than the sham control group, mostly likely because of incomplete denervation. The immunohistologic staining on autopsy demonstrated both higher sympathetic and parasympathetic nerve density in the ablation arm but not the sham group. Pulsed-field ablation (PFA) is a novel ablation technology that induces loss of function in arrhythmogenic myocardial tissue while sparing adjacent tissues and nerves (and therefore GPs). Acute VRs are seen frequently during PFA energy delivery in as many as 33% of patients [81]. In a recently published study, Guo et al. [82] evaluated the effect of PFA PVI on the ANS. Despite the frequent occurrence of VR during PFA applications, serum nerve injury biomarkers did not show any changes immediately post-ablation and 24 h after ablation. Furthermore, the HR variability did not differ post-ablation. Thus, the VRs appear to be a neurological stress response due to electrical stimulation rather than representative of nerve injury, per se. Because the preliminary experience with PFA suggests comparable effectiveness with other ablative methods, the additional benefit of modifying the ANS remains uncertain.

## 9. Conclusions

Experimental and clinical studies suggest that the intrinsic cardiac ANS plays an important role in both the initiation and the maintenance of AF. The ablation of GP may be considered adjunctive to PVI for preventing PV firing and potentially reducing AF recurrence. Although the data thus far have been mixed supporting GP ablation, randomized controlled studies have shown promise for therapies such as additional GP ablation and VOM ethanol ablation. Additional studies with larger randomized studies are needed.

## Figures and Tables

**Figure 1 life-13-00383-f001:**
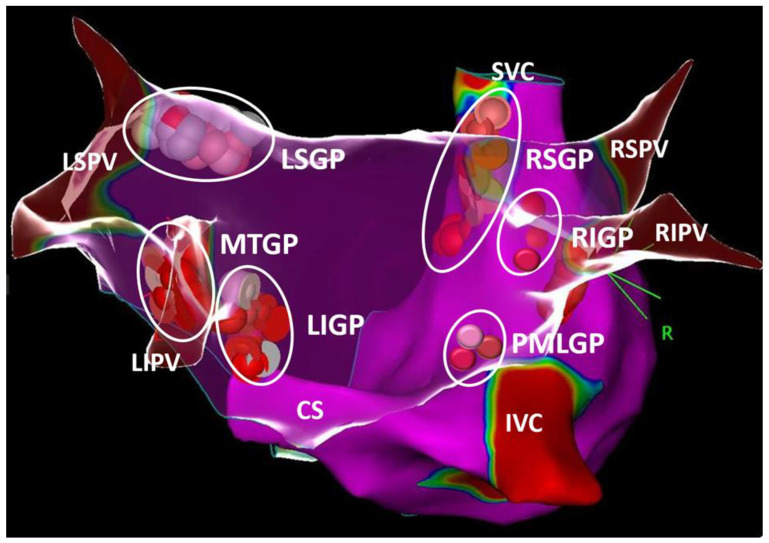
The schematic view of ganglionated plexi (GPs). White, pink, and red dots show distribution of ablation points based on fragmented bipolar electrograms. Please see text for details. CS, the coronary sinus; IVC, the inferior vena cava; LIGP, the inferior left atrial GP; LIPV, the left inferior pulmonary vein; LSGP, the superior left atrial GP; LSPV, the left superior pulmonary vein; MTGP, the Marshall tract GP; PMLGP, the posteromedial left atrial GP; RIGP, the inferior right atrial GP, RIPV, the right inferior pulmonary vein; RSGP, the superior right atrial GP; RSPV, the right superior pulmonary vein; SVC, the superior vena cava.

**Figure 2 life-13-00383-f002:**
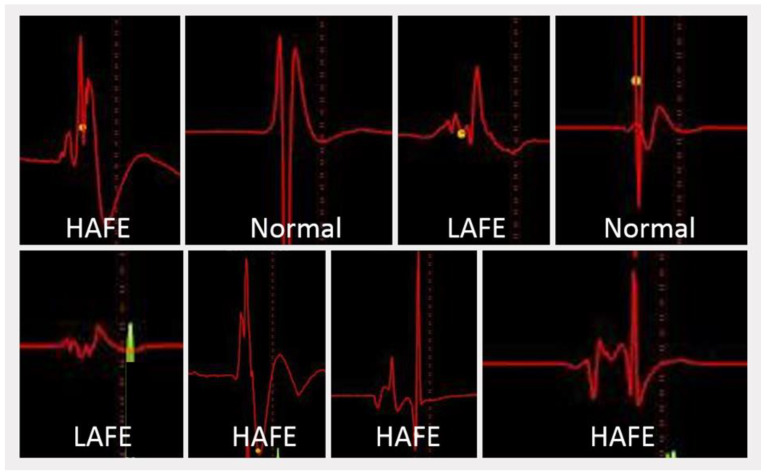
Three bipolar atrial electrogram types for ganglionated plexus mapping. HAFE, high amplitude fragmented electrogram; LAFE, low amplitude fragmented electrogram; Normal, normal atrial electrogram.

**Figure 3 life-13-00383-f003:**
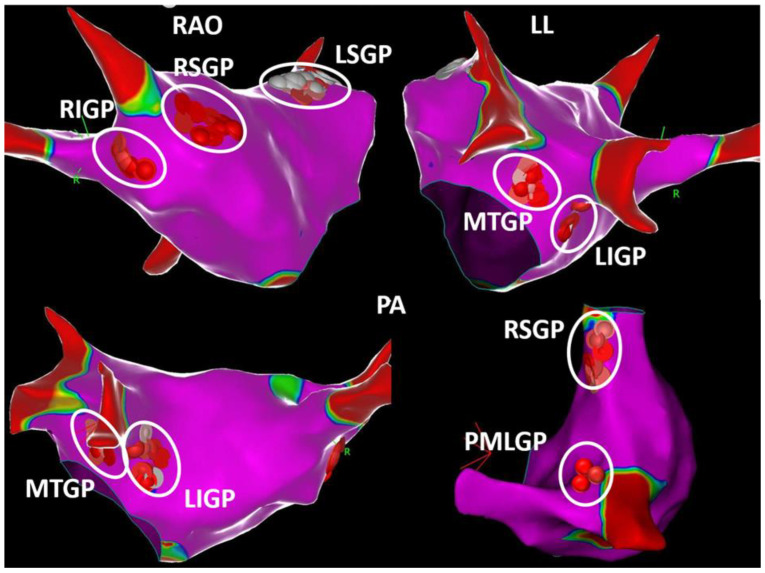
Distribution of ablation points in different views. White, pink, and red dots show distribution of ablation points based on fragmented bipolar electrograms. Please see Figure 1 for abbreviations: LL, left lateral; PA, posteroanterior; RAO, right anterior oblique.

**Table 1 life-13-00383-t001:** Advantages and disadvantages of different ganglionated plexus mapping techniques.

Technique	Setting	Advantages	Disadvantages
Spectral analysis [16,40]	Time to amplitude-based electrograms are converted to frequency spectrum through the fast Fourier transforms The software (Pachón-TEB2002) works with a customized 32-channel polygraph Fibrillar potentials (AF nests) with a highly fragmented, heterogeneous, and right-skewed spectral distribution show vagal innervation sites	Depending on the filters applied during the recordings, all desired frequency spectra can be studied A concise method for the qualitative and quantitative assessment of the proportion of high-frequency components within atrial electrogram	No integration with current 3D mapping systems No chance for online analysis of electrogram
High frequency Stimulation (HFS)	20 Hz, 10–30 V, and pulse width of 1–10 ms, for 2–5 s [41,42] 20 Hz, 10 V, up to 10 s to detect the atrioventricular dissociating-GPs and 40 Hz, 10 V for 80 ms to detect the ectopy-triggering-GPs [19,43] A positive vagal response, which is defined as a transient sinus bradycardia or atrioventricular block, shows vagal innervation sites	A well-studied method on both atrial fibrillation ablation and cardioneuroablation [19,41,42,43,44,45] Can be used to evaluate vagal denervation in each GP site	Need stimulators with high-frequency capability (e.g., Grass stimulator S-88 and Micropace EPS320) No consensus for proper protocol No consensus for definition of positive vagal response Conscious patients may not tolerate more than 15 V and general anesthesia is usually needed Induction of atrial fibrillation HFS-based strategy has not demonstrated an advantage over empirical anatomic ablation [46] Vagal response characteristics can be differentially affected by conscious and sedation deep sedation [47]
Electrogram analysis	Bipolar endocardial atrial electrograms are evaluated for number of deflections at filter settings of 200–500 Hz [45] * The electrograms demonstrating greater or equal to 3–4 deflections in regions which are anatomically consistent with GP sites are tagged as ablation targets [46,47,48,49]	A readily available method for the semi-quantitative assessment of high-frequency components (can be considered a simplified variant of spectral analysis) does not require any specific technology and can be used in all electrophysiology centers No extra costs Related to shorter procedure time than combination of spectral analysis and high-frequency stimulation with a similar success rate [45] Reproducible by first-time operators [50]	The fragmented electrograms may also be found at the sites with complex architecture of atrial myocardium, (e.g., at the pulmonary veins ostia, at ridges, multiple layers (coronary sinus), and most likely at the interatrial septum sites where GPs important for cardioneuroablation are also located The assessment might be less concise compared to spectral analysis Fibrosis may also cause fragmentation due to slow conduction Decisions made by humans based on visualizations of data may demonstrate low reproducibility
Anatomical Approach	The technique can be used as adjunctive to spectral analysis, high-frequency stimulation, or fragmented electrograms; or as a stand-alone strategy [46,51,52] Empirical ablation is performed in the presumed areas of GPs [53]	No extra costs does not require any specific technology and can be used in all electrophysiology centers Easily applicable Better results than high-frequency stimulation-based approach [46]	No definition for ablation target No definition for ablation endpoint Reproducibility by first-time operators?
Computed Tomography (CT) imaging		It might be used to define posteroseptal part of the superior vena cava to avoid transseptal puncture if it is properly performed and correctly merged with an electroanatomical map [54] It may increase the confidence in performing left atrial or bi-atrial cardioneuroablation, in case of complex and variable pulmonary vein and left atrium anatomy It may disclose major anatomical abnormalities that could interfere with successful cardioneuroablation	The benefit of registering the CT anatomic images to the electroanatomical map has not been reliably proven for the success of cardioneuroablation as well as for other complex ablation procedures for cardiac arrhythmias. It might be associated with non-necessary additional costs. Another concern in young patients is the radiation exposure which is multi-fold higher than that during cardioneuroablation itself that can be performed in a near-to-zero fluoro manner. Improper use of image registration may be misleading in non-experienced hands.
123I-metaiodobenzylguanidine (mIBG)	Imaging is performed following injection of 123I-mIBG on a dedicated cardiac solid-state SPECT camera Images are acquired for 20 min with the region of interest [55] All patients undergo cardiac CT After manual corrections to the segmentation, a representative 3D surface mesh file is created for each chamber that is then used for co-registration with SPECT tomograms	mIBG LA uptake areas are usually correlated with HFS [55] Left atrial innervation imaging by SPECT may integrate with invasive HFS and provide the planning of the ablation procedure	Mapping parameters are not standardized the technique needs validation against High-frequency stimulation or visual electrogram analysis No integration with current 3D mapping systems

* This setting is available in WorkMate Claris system (St Jude Medical, Abbott, St. Paul, MN, USA). In Prucka Cardiolab system (GE Healthcare, Wauwatosa, WI), 100–500 Hz can be selected as default setting [40].AF, atrial fibrillation; GP, ganglionated plexus.

**Table 2 life-13-00383-t002:** Summary of randomized-controlled studies investigating the role of autonomic nervous system ablation in patients with atrial fibrillation.

Author	Patient Number	Ablation Route	GP Localization	Population	Randomized Groups	Outcomes (AF/AT Free Survival)
Pokushalov [52]	80	Endo	HFS or Anatomical	PAF	HFS Anatomical	42.5% in HFS 77.5% in anatomical (*p* = 0.019)
Katritsis [53]	67	Endo	Anatomical	PAF	GP + PVI PVI	85.3% in GP + PVI 60.6% in PVI (*p* = 0.019)
Katritsis [18]	242	Endo	Anatomical	PAF	PVI GP GP + PVI	56% in PVI 48% in GP 74% in GP + PVI (*p* = 0.004)
Mamchur [73]	120	Endo	Anatomical	Per AF/ LSPAF	GP PVI GP + PVI	38% in GP, 56% in PVI 69% in GP + PVI (*p* = 0.006 for GP + PVI vs GP)
Pokushalov [72]	264	Endo	Anatomical	Per AF/LSPAF	PVI + LL PVI + GP	47% PVI + LL 54% (PVI + GP) (*p* = 0.29)
Berger [70]	240	Thora	HFS+ anatomical	PAF/ Per AF	GP + PVI PVI	55.6% in GP + PVI 56.1% in PVI (*p* = 0.407)
Sandler [43]	67	Endo	HFS	PAF	GP PVI	49% in GP 61% in PVI (*p* = 0.27)
Kim [19]	102	Endo	HFS	PAF	GP PVI	50% in GP 64% in PVI (*p* = 0.09)

GP, ganglionated plexus; HFS, high-frequency stimulation; LL, linear ablation; LSPAF, long-standing persistent atrial fibrillation, paroxysmal atrial fibrillation; Per AF, persistent atrial fibrillation; PVI, pulmonary vein isolation.

## Data Availability

Not applicable.

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
