# Peer review of "Intrinsic Cardiac Neuromodulation in the Management of Atrial Fibrillation- A Potential Missing Link?"

_life, 2023, doi:10.3390/life13020383_

Round 1

Reviewer 1 Report

The review is well written in terms of content. However, typos need to be corrected.

I have no major comments. However, I would like the authors to refer to pulse field ablation (PFA) in the discussion section. PFA as a method of AF ablation will be more and more widely used. By definition, PFA does not damage nerves (and therefore GPs) and adjacent tissues, and its effectiveness in the treatment of AF is comparable to other ablative methods. Is it therefore necessary to modify the ANS since the lack of modulation during PFA shows comparable effectiveness. Please refer to this.

Author Response

Comments and Suggestions for Authors

The review is well written in terms of content. However, typos need to be corrected.

Author response: We would like to thank to the reviewer to point out this important issue. All the text were edited for grammatical and spelling errors.

I have no major comments. However, I would like the authors to refer to pulse field ablation (PFA) in the discussion section. PFA as a method of AF ablation will be more and more widely used. By definition, PFA does not damage nerves (and therefore GPs) and adjacent tissues, and its effectiveness in the treatment of AF is comparable to other ablative methods. Is it therefore necessary to modify the ANS since the lack of modulation during PFA shows comparable effectiveness. Please refer to this.

Author response: We would like to thank to the reviewer to point out this important issue. We have added PFA in the “Unanswered Questions” section. The text has been highlighted with yellow color.

Reviewer 2 Report

The paper extensively deals  the subject. It would be better in my opinion not too long sentences and shorter periods.

Author Response

Reviewer 2

Comments and Suggestions for Authors

The paper extensively deals the subject. It would be better in my opinion not too long sentences and shorter periods.

Author response: We would like to thank to the reviewer to point out this important issue. All the text were edited based on the reviewer’s suggestions.